# The Study of Aircraft Accidents Causes by Computer Simulations

**Paweł Szczepaniak** [1,*] **, Grzegorz Jastrzębski** [1,*] **, Krzysztof Sibilski** [1] **and Andrzej Bartosiewicz** [2]

1   Air Force Institute of Technology, Księcia Bolesława 6, 01-494 Warsaw, Poland; krzysztof.sibilski@itwl.pl
2   State Commission on Aircraft Accidents Investigation, Chałubińskiego 4/6, 00-928 Warsaw, Poland; andrzej.bartosiewicz@mi.gov.pl
*   Correspondence: pawel.szczepaniak@itwl.pl (P.S.); grzegorz.jastrzebski@itwl.pl (G.J.)

**Abstract:** Defects in an aircraft can be caused by design flaw, manufacturer flaw or wear and tear from use. Although inspections are performed on the airplane before and after flights, accidents still result from faulty equipment and malfunctioning components. Determining the causes of an aircraft accident is an outcome of a very laborious and often very long investigation process. According to the statistics, currently the human factor has the biggest share within the causal groups. Along with the development of aviation technology came a decline in the number of accidents caused by failures or malfunctions, though such still happen, especially considering aging aircraft. Discovering causes and factors behind an aircraft accident is of crucial significance from the perspective of improving aircraft operational safety. Effective prevention is the basic measure of raising the aircraft reliability level, and the safety-related guidelines must be developed based on verified facts, reliable analysis and logical conclusions. This article presents simulation tests carried out by finite element method and constitutive laboratory tests leading to the explanation of the direct cause of a military aircraft accident. Computer-based simulation methods are particularly useful when it comes to analysing the kinematics of mechanisms and potential stress concentration points. Using computer models enables analysing an individual element failure process, identifying their sequence and locating their primary failure source.

**Keywords:** threaded connection; computer simulations; aviation safety; aircraft accidents

## 1. Introduction

An aircraft is an assemblage of complex and highly integrated sub-systems. It is generally agreed that there exist certain precursors to each accident and incident. If one of these precursors is not recognized and the underlying condition that has caused it is not corrected in time, then it can graduate into an incident or even an accident. Aircrafts are highly engineered systems, endowed with redundancies and fail-safe features [1].

Causal analysis of an accident or incident seeks to establish those factors that were judged to be directly responsible in causing the event (primary causal factors) and those that contributed to the event (secondary causal factors) by deconstructing the accident. For these causal factors, a causal chain can usually be established for each accident or incident [2]. The advantage of causal chain analysis is that in the case of multiple causes and multiple accidents or incidents, the common events or elements in the chain can be identified and subjected to greatest attention. Thus, the safety system can concentrate on those common events and maximize its responsiveness and effectiveness in cutting down-times, and reducing or eliminating accidents. The perceived disadvantage of this approach is that it is reactive rather than proactive. That is, the regulating authority and the industry (or the

military operators) seek to eliminate the causal factor after the accident in order to prevent accidents due to the same cause from happening again [2–4].

One of the commonly used disaster models is the "Swiss cheese" model proposed by Reason. In the Swiss cheese model, an organisation's defences against failure are modelled as a series of barriers, represented as slices of the cheese. The holes in the cheese slices represent individual weaknesses in individual parts of the system, and are continually varying in size and position in all slices. The system, as a whole, produces failures when holes in all of the slices momentarily align, permitting "a trajectory of accident opportunity", so that a hazard passes through holes in all of the defences, leading to an accident [5,6].

An example showing how a seemingly minor defect caused by a disaster can be the Trans World Airlines Flight 800 (TWA 800) airliner crash. This accident is also a good example of the Swiss cheese disaster model. TWA 800 was a Boeing 747-100 that exploded and crashed into the Atlantic Ocean near East Moriches, New York, on 17 July 1996. The four-year National Transportation Safety Board (NTSB) investigation concluded with the approval of the Aircraft Accident Report on 23 August 2000, ending the most extensive, complex and costly air disaster investigation in United States history to that time [7]. The report's conclusion was that the probable cause of the accident was explosion of flammable fuel vapours in the centre fuel tank. Although it could not be determined with certainty, the likely ignition source was a short circuit. Problems with the aircraft's wiring were found, including evidence of arcing in the Fuel Quantity Indication System (FQIS) wiring that enters the tank. The FQIS on Flight 800 is known to have been malfunctioning; the captain remarked on what he called "crazy" readings from the system approximately two minutes and thirty seconds before the aircraft exploded. As a result of the investigation, new requirements were developed for aircrafts to prevent future fuel tank explosions. The NTCB's final opinion is: "An explosion of the center wing fuel tank (CWT), resulting from ignition of the flammable fuel/air mixture in the tank. The source of ignition energy for the explosion could not be determined with certainty, but, of the sources evaluated by the investigation, the most likely was a short circuit outside of the CWT that allowed excessive voltage to enter it through electrical wiring associated with the fuel quantity indication system." [5,8].

This article presents the results of finite element method (FEM) simulations and laboratory tests leading to the detection of the "primary cause" the aircraft crash. The term primary cause, defined as the most critical cause factor associated with a particular accident, can be deceiving and is often subject to interpretation.

Studying the causes of an aircraft accident is usually an outcome of a very arduous and often long investigation process. The aircraft reliability degree, apart from structural safety, is also significantly affected by the quality of operation [9,10]. This applies to all work, both repairs as well as ongoing maintenance. As empirical data have shown, often a seemingly insignificant shortcoming may lead to catastrophic outcomes [9]. Based on the effects of an event, as a result of a thorough study utilizing advanced numerical methods, it is possible to accurately diagnose the cause of the event, which will contribute to effective prevention and possibly improve flight safety. Studies based on advanced numerical techniques are now widely used in numerous fields of industry, the aerospace, aviation and automotive industries in particular. In general, numerical methods are used for the calculations in terms of machine and assembly parts at the engineering stage or when optimizing existing structures [11–13]. The numerous research work conducted at the Air Force Institute of Technology (AFIT) involved computer simulation research regarding the elements of an aircraft landing gear mechanism (Figure 1a), aircraft hydraulic lines (Figure 1b), threaded connections and an aircraft variable displacement hydraulic pump. A Computer Aid Design 3 Dimensional (CAD3D) application, which contains a feature to study mechanism kinematics was used to recreate the geometry of unit parts and assemblies. The study also involved using 3D object scanning techniques aimed at recreating the complex geometry. The finite element method (FEM) analysis was used for the strength tests. Assemblies and elements can be subjected to various numerical tests—structural strength, flow and thermal analyses [14–17]. Nonetheless, it should be noted that numeric methods are not

perfect, and their application for analysing technical issues is not easy and requires experience in this field. The entirety of this publication is closely devoted to the issues of modelling the strength of a threaded connection.

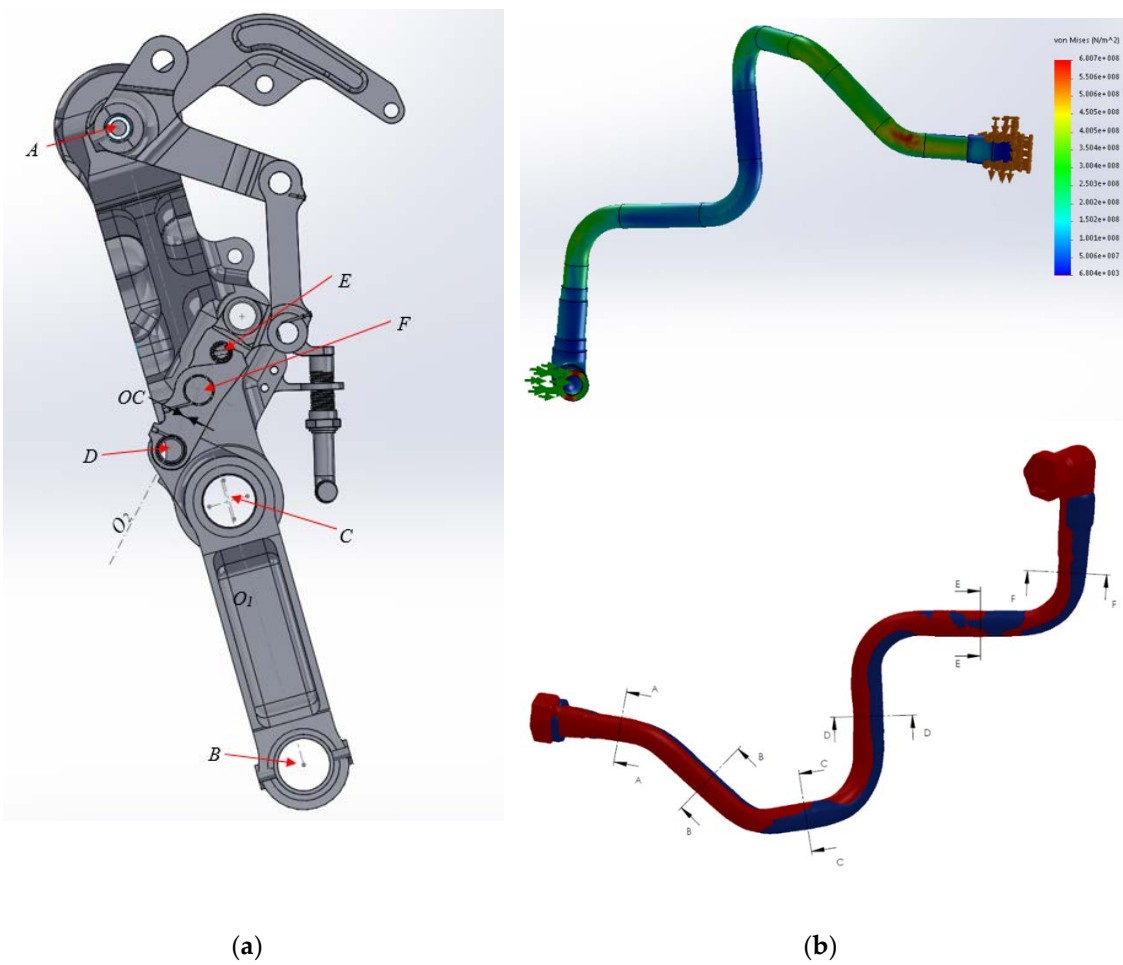

(**a**)　　　　　　　　　　　　　　　　　　　　　　(**b**)

**Figure 1.** Examples of applying simulation methods in investigating air accident causes: (**a**) aircraft retractable landing gear strut kinematics, (**b**) stress concentration areas and the geometry of an aircraft hydraulic line.

Other issues concerning computer-based tests involving the landing gear, rigid hydraulic lines and the pump will be addressed in further articles within this monographic cycle. This publication presents detailed results of the numerical and experimental tests involving a shank with a thread and cooperating nut, which are the elements of an aircraft fuel system assembly (Figure 2). The various aspects of numerical analyses of threaded connection are approached in scientific research [18–22]. Whereas the authors have not encountered a strength analysis of a threaded connection with modelled operational wear and tear in the source literature. These studies are to confirm or contradict the thesis that, at a given wear level of this connection, and under its strictly defined load, its load-bearing capacity may be lost, resulting in the loss of material continuity. The conducted simulations involved a conventionally new thread and with modelled wear, which was recreated based on the geometry of an actual object. The article presents results only for a thread with a modelled wear. Apart from the simulations, the study also involved conducting experiments, the results of which were used to verify the developed model parameters and simulations conditions.

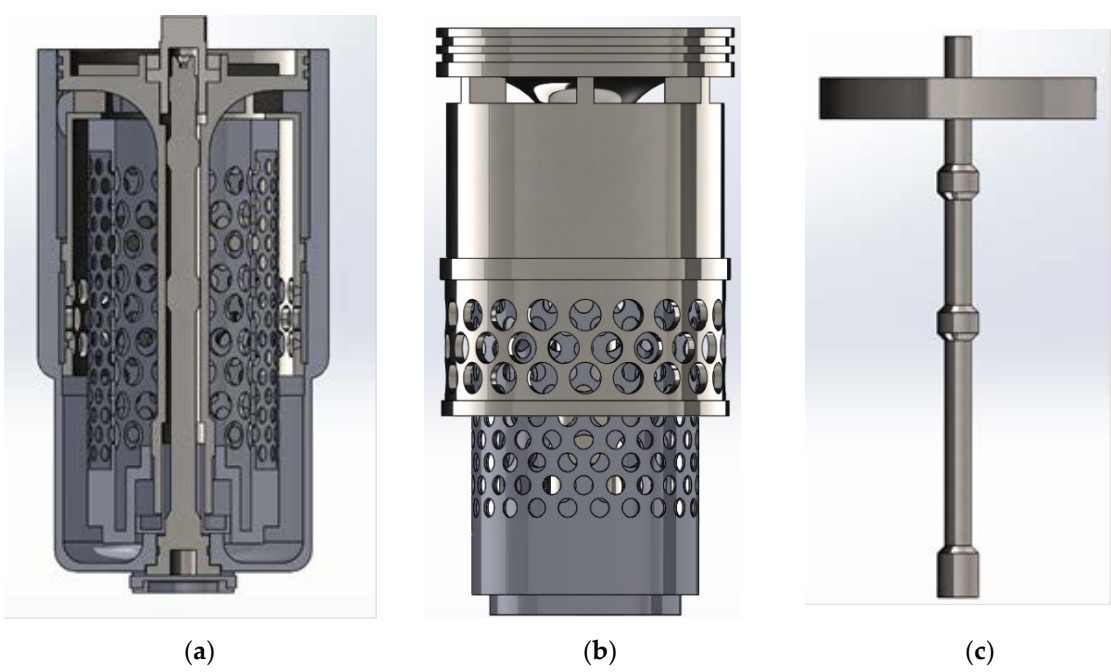

| (**a**) | (**b**) | (**c**) |

**Figure 2.** A computer solid model of the studied fuel system assembly: (**a**) cross-section, (**b**) view without housing, (**c**) simplified model for strength analysis.

## 2. Simulations and Experimental Tests

The first stage of the study involved mapping a specific thread within a solid model, which was conventionally new, with the parameters set out in the literature. A seven-pitch M14x1.5 thread was subjected to the tests. This model was then modified in terms of the pitches on the shank (Figure 3). The modification involved modelling the wear of shank thread pitch, based on the measurements of an actually worn thread. The developed CAD model was subjected to discretisation. The created numerical model does not reflect thread pitch microdamage and the contact surfaces are "numerically" smooth, which is a sort of a simplification. A set of "non-penetration" contact assembly functions were used for modelling the joint. Developing a discrete microstructure in the form of a microscopic numerical model is not an easy task, but in some cases can be of crucial importance. An example of such an approach in reality is presented by the authors of the analysis involving the numerical tests of braking system elements [18,21].

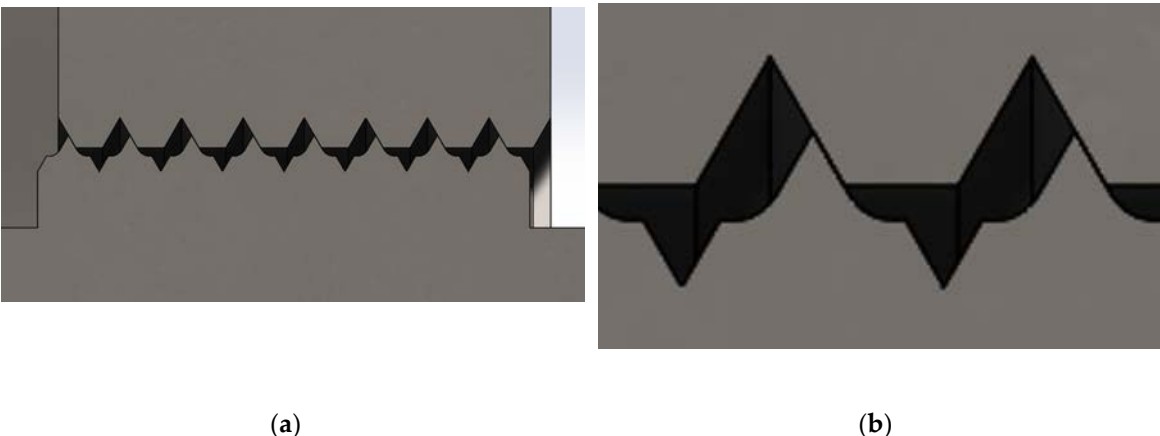

| (**a**) | (**b**) |

**Figure 3.** View of a thread contour with modelled shank wear: (**a**) all pitches, (**b**) two pitches magnified.

The material used to construct the shank of the filter with the studied thread is the Ti8Al1Mo1V titanium alloy with the short-term strength of 937 MPa, yield point of 910 MPa and Young's modulus

of 120 GPa. The modelled nut is made of 30H2N2M alloy steel. The calculations adopted a short-term strength of 723 MPa, yield point of 620 MPa and Young's modulus of 210 GPa [23].

The discrete model (Figure 4a) has green arrows marking the fastening points (restraint) and red arrows marking the load. The construction of a spatial discrete model was based on solid elements in the shape of tetrahedrons, with a maximum size of approx. 1.6 mm. Grid refinements within the area of interest were developed using grid controls, four Jacobian points, with the minimum element size of approx. 0.06 mm. The total number of nodes was 1,230,607, whereas the total number of elements was 825,131. The share of elements with a shape factor below 3 was around 95%. There were no identified distorted elements. The numerical tests utilized the "h" optimization method and resulted in the development of a convergence graph aimed at increasing the accuracy of the obtained results. The optimization criterion was the resultant stress value, as per the Huber–Mises hypothesis, and the deformation energy error was specifically relative. The post-optimization energy error was below 6%, which demonstrates satisfactory compliance of the results according to the data. The general dependence on Huber–Mises reduced stress $\sigma_{red}$ [23–27]:

$$\sigma_{red} = \sqrt{\frac{\left(\sigma_x - \sigma_y\right)^2 + \left(\sigma_y - \sigma_z\right)^2 + \left(\sigma_z - \sigma_x\right)^2}{2} + 3\left(\tau_{xy}{}^2 + \tau_{yz}{}^2 + \tau_{xz}{}^2\right)} \tag{1}$$

where: $\sigma_{x(y,z)}$—normal stress towards the x (y, z) axis of the coordinate system, $\tau_{xy(yz,xz)}$—shearing stress within the xy (yz, xz) plane cross-section.

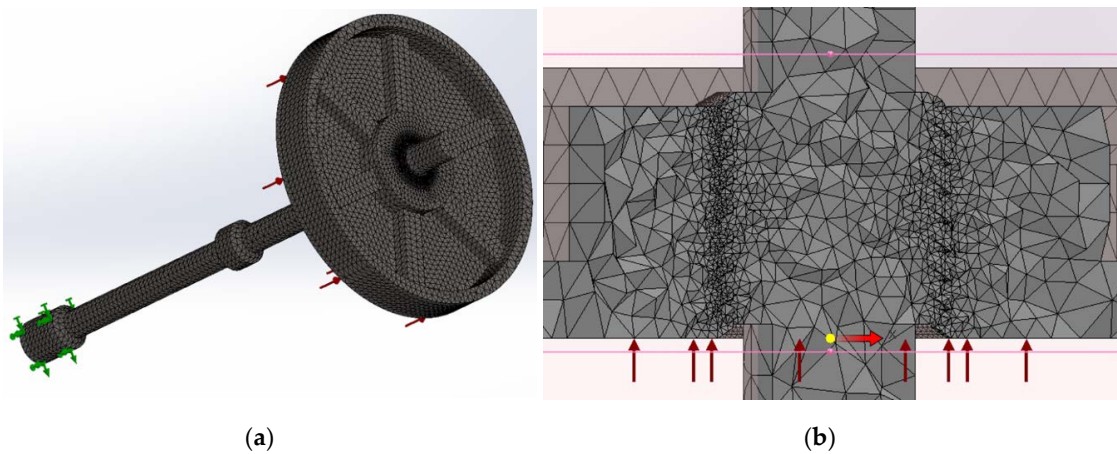

| (a) | (b) |

**Figure 4.** Solid model grid for the studied fuel system assembly: (**a**) complete discrete model, (**b**) within the studied thread-refinement.

The numerical strength tests using the hypothesis on the highest shape deformation energy are mainly utilized to research and calculate elements within the elastic range [13,27,28]. The phenomenon of material failure under stress is much more complex than just the appearance of the first permanent deformations [27]. Parting failure is caused by stress resulting in overcoming of the cohesion forces between material particles. In the case of calculating material failure conditions, it is recommended to apply Mohr's hypothesis. According to the Mises model, the limit of elasticity and the yield point are of the same value. It is assumed that reaching this value, hence the beginning of plastic deformation, equals the loss of thread's load-bearing capacity [27]. Therefore, strength hypotheses for ductile materials are very often called the yield conditions. Given the auxiliary nature of the numerical calculations in terms of solving engineering issues, the authors believe that applying a universal reduced stress model is sufficient. Other authors have also used the model as per the Huber–Mises hypothesis within their numerical studies involving a threaded connection.

The highest stress values (Figure 5a,b and Figure 6a,b) were obtained as expected, at the interface point between the elements of the modelled, worn threaded connection. The simulation conditions

assumed loading the filter cover element with a static pressure of 8.5 MPa. Under such load, very large areas of the modelled thread, the first three pitches experience significantly exceeded yield point value.

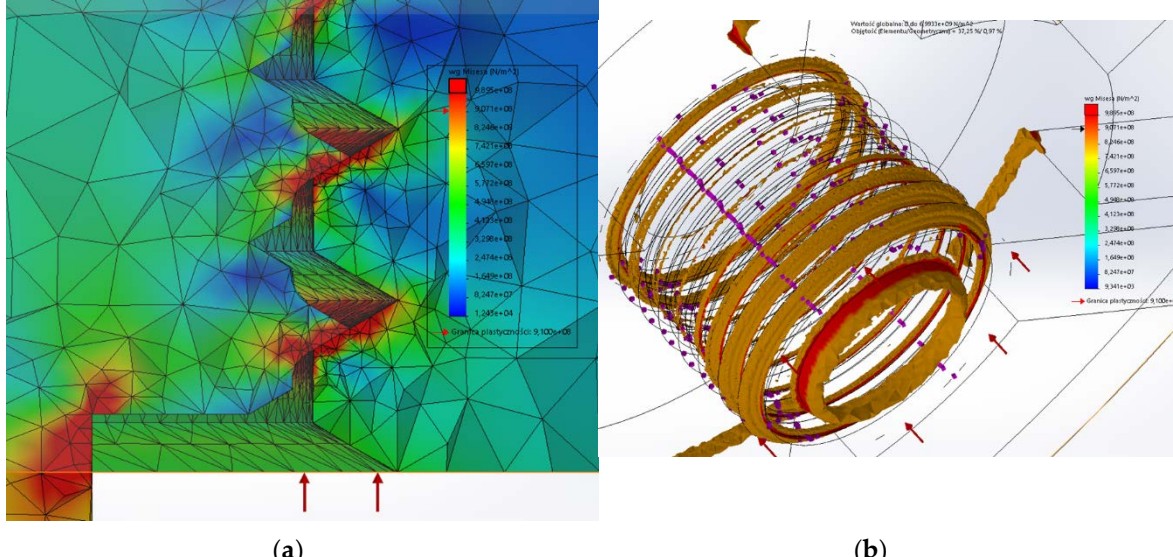

(**a**)　　　　　　　　　　　　　　　　　　　　　　　　　　(**b**)

**Figure 5.** The stress charts for the studied shank thread with simulated wear and experimental failure load: (**a**) graph cross-section with a grid (first 2 are the most loaded pitches), (**b**) separated area with a clearly exceeded material yield point.

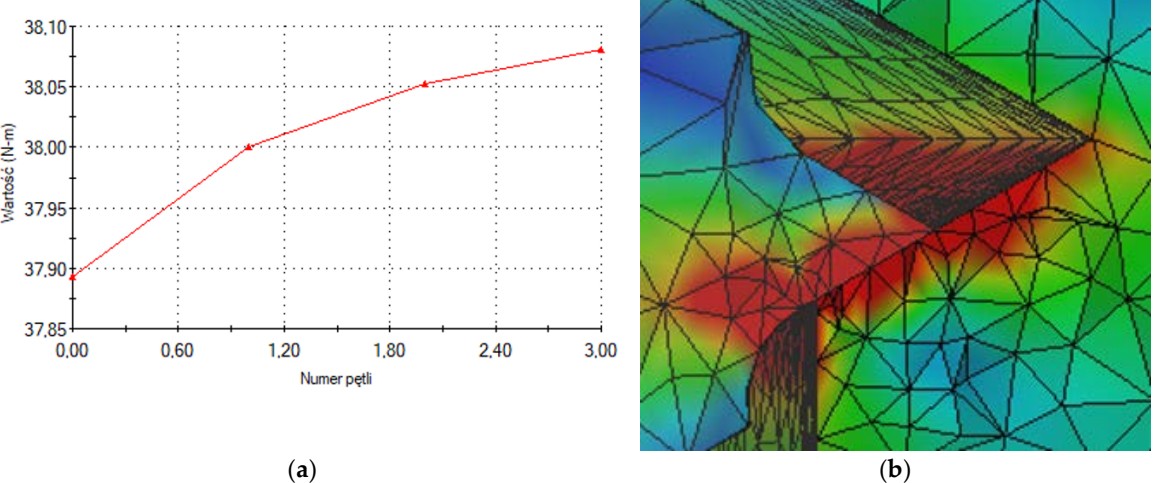

(**a**)　　　　　　　　　　　　　　　　　　　　　　　　　　(**b**)

**Figure 6.** The outcomes of the optimization algorithm as per method h: (**a**) convergence graph, (**b**) reduced stress distribution, cross-section, magnification, with a grid for the first thread pitch.

Such a situation may confirm the load capacity loss of a thread with such material properties, geometry, diagram and load value. The results of numerical tests enabled also to determine the features of the experimental test bench. The experimental tests were conducted for two threaded connections with a similar operational wear level. It has been developed test methodology, using a specially designed and constructed test bench (diagram—Figure 7). The test liquid, with a similar pressure value, resulted in the loss of the threaded connection's load capacity and, ultimately, in the "shearing" of thread pitches and the loss of tightness within the studied assembly. The experimental test results are shown in Figure 8.

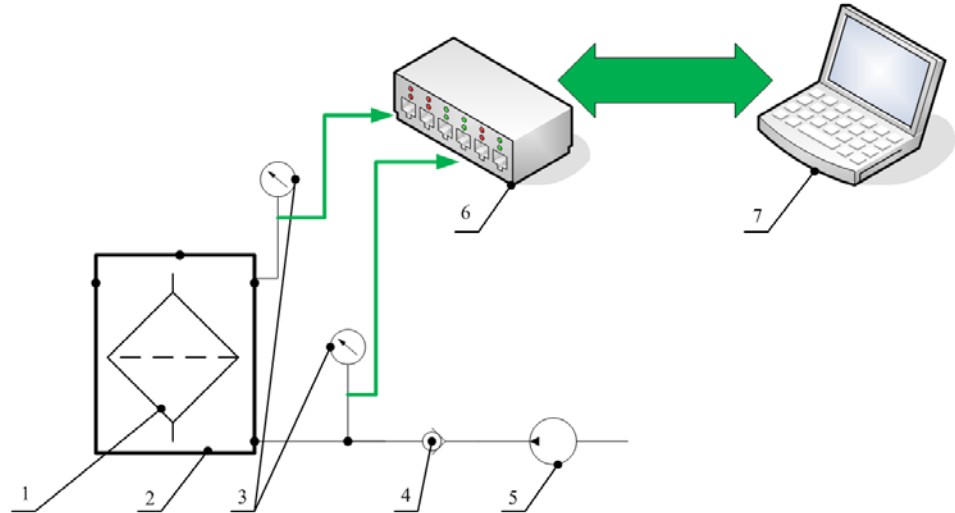

**Figure 7.** Diagram of a test bench for static loading of an aircraft fuel filter assembly: 1—filter, 2—NC filter body, 3—pressure sensors, 4—one-way valve, 5—manual hydraulic pump, 6—analogue signal translator, 7—computer.

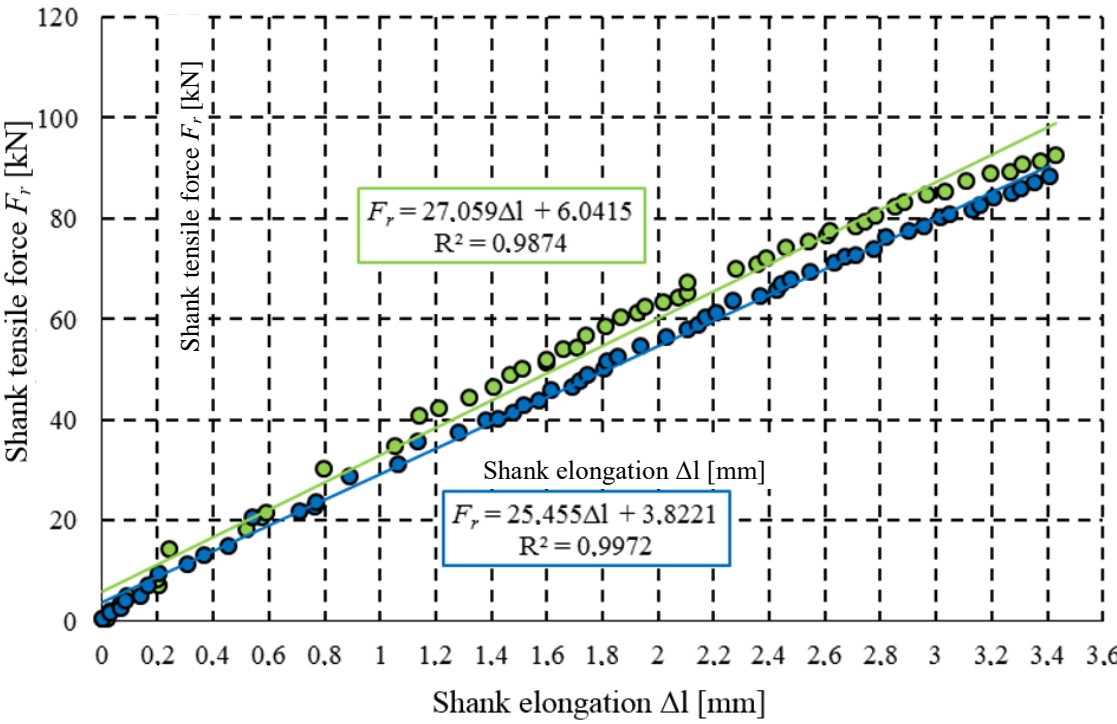

**Figure 8.** The results of experimental tests involving rupturing threads of two shanks.

The obtained value of the threaded connection destructive force is 84 and 86 kN (Figure 8) for shank No. 1 and 2, respectively, which corresponds to the value of the hydrostatic pressure inside the assembly housing at a level of 8.2–8.3 MPa. Shank elongation confirmed by experimental tests, relative to the tensile force (increase of internal pressure within the fuel assembly) is of linear and flexible character, in accordance with Hooke's law [27]—$R^2$ coefficient of approximately 0.99 and 1.00.

With the elongation value within the elastic range at a level of 3.4 mm, there is a sudden loss of thread load-bearing capacity, pitch rupturing and a simultaneous loss of assembly tightness.

### 3. Conclusions

Many aircrafts operated in the military reach a state referred to as aging at continuous operational service. That is why military aircrafts are in constant use for many decades, maintaining satisfactory operational capabilities. Aging of an aircraft is not the same as it becoming obsolete. The time when an aircraft reaches the aging state is usually much more difficult to determine. It is important to distinguish between the characteristics of the structure of a young aircraft and an aging aircraft. Sustainment of an aircraft is the act of keeping it operational (i.e., airworthy). Maintenance of an aircraft (that is, the work done by mechanics in keeping it airworthy) is one aspect of sustainment. However, sustainment also includes the engineering analyses and tests needed to determine an adequate maintenance plan for the aircraft. Sustainment is life management. One task of sustainment is the determination of structural inspections based on damage tolerance principles. These inspections protect against failure from defects that could be in the structure because of manufacturing or from operational service [29].

In the absence of support from the aircraft manufacturer and the lack of original technical documentation, it was necessary to reproduce those documentations by reverse engineering methods. Such works was carried out at the Air Force Institute of Technology. These works allowed the maintenance of continuous operation of post-Soviet combat aircraft in the Polish Air Force.

Unlike commercial aircraft structures, military aircraft airframe structures are rather fatigue resistant. Therefore, in the case of combat aircraft, disasters caused by fatigue of the airframe structures are rare. More frequent causes of combat aircraft disasters are engine damage or malfunctions of seemingly insignificant aircraft subsystems and damages of equipment elements. This article was devoted to the study of a fighter aircraft crash. The use of the finite element method allowed the detection of the cause of damage in the fuel system, which was the direct cause of the accident.

The first step of this case study was developing of the geometrical model of the threaded connection. This geometrical model was based on the measurements obtained in respect of an actual object. The formulated boundary conditions (load, fastening) allowed to conduct numerical calculations for a geometrical model with recreated operational wear. The results of numerical tests confronted with environmental studies can be used as a base to evaluate the actual load-bearing capacity of a threaded connection for a given level of operational wear. At the same time, the experimental test results enabled evaluating and validating the developed numerical method.

Based on the results of the numerical tests, it is possible to estimate a feasible load-bearing capacity of actual, operated threaded connections within a studied fuel system assembly. Using a computer simulation method enabled to verify the validity of the thesis in the analysed case. The Huber–Mises reduced stress values $\sigma_{red}$ in the interfacing areas of the shank thread pitches and an aircraft fuel assembly nut, with modelled operational wear, obtained in the course of the simulation tests are consistent with the experimental results. With the assumed internal fuel load of 8.5 MPa over significant areas of the studied thread, the reduced stresses $\sigma_{red}$ exceed the yield point value. Such a state of the stress is a piece of evidence of lost load-bearing capacity of the threaded connection, which was confirmed by the results of experimental tests.

There are works devoted to forensic engineering of the causes of aviation accidents focused on reconstruction of flight trajectories [30,31]. Simulation tests of flight dynamics in pre-disaster time intervals are particularly important in the absence of sufficient information registered by Flight Data Recorder (FDR) [6,32]. In cases of full registration of flight parameters by FDR, simulation reconstruction of flight parameters is generally not necessary. However, questions arise regarding the impact and damage of aircraft components to the occurrence of a disaster.

This article's objective was to assist the Air Force in making difficult yet necessary choices regarding its aging fleets in order to fulfil national security objectives at lowest cost. We have shown the usefulness of computer methods in the process of reaching to indicate the direct cause of an aircraft crash. It is an example of the use of computer simulation methods in forensic engineering. A simulation study of the cause of a fighter aircraft crash was carried out. The use of the finite element method allowed the detection of the cause of the fuel system failure, which was the direct cause of the accident.

**Author Contributions:** Conceptualization, P.S., A.B. and K.S.; methodology, P.S. and G.J.; software, P.S. and G.J.; validation, P.S., G.J. and A.B.; formal analysis, P.S. and K.S.; investigation, P.S., G.J., and A.B.; resources, P.S.; data curation, P.S., G.J. and A.B.; writing—original draft preparation, P.S. and K.S.; writing—review and editing, K.S.; supervision, P.S. and K.S.; funding acquisition, P.S. All authors have read and agreed to the published version of the manuscript.

**Funding:** This research was internally funded by the Air Force Institute of Technology.

**Conflicts of Interest:** The authors declare no conflict of interest.

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
