# Peer review of "The Study of Aircraft Accidents Causes by Computer Simulations"

_aerospace, doi:10.3390/aerospace7040041_

Round 1

Reviewer 1 Report

Please see comments in the attached file.

Author Response

Reviewer 1

Thank you very much for the thorough review.

According to the comments:
we've added authors' email addresses and information about the authors,
we have shortened the summary and have added explanations of abbreviations.

Reviewer 2 Report

  1. Abstract - the abstract does not summarise this research, authors do not talk about the gaps in similar work, how they addressed the gaps, no conclusion on the proposed solution effectiveness, abbreviations not fully spelled, too lengthy sentences which end up confusing the readers, some irrelevant point e.g. 
  2. Introduction - the introduction talks about numerical tests for aircraft elements. At this stage, not sure what research objectives of this paper. Content of the Introduction does not reflect the title of the paper. The authors need to know that occurrences involving aircraft are not only due to aircraft mechanical failure. The a numerous factors which is not mentioned in this paper. I suggest that the Authors refine their research topic according to the scope of their work.
  3. The methodology is not defined.
  4. The Simulation and test section does not reflect the scope of the paper based on the title.
  5. Language has to be improved.
  6. Too many abbreviations not spelled in full.

Author Response

Reviewer 2

Thank you very much for the thorough review.

Ad. 1

We shortened the summary and explained the purpose of the research more precisely.

Ad 2

In the introduction, a paragraph has been added that precisely specifies the study cycle

Ad 3.

The methodology is defined. The first: numerical tests (section 86-159), and the second: experimental tests (section 160-186). Due to the special scope of this work concerned research of a fighter aircraft crash, not all detailed information about the research could be published.

Ad 4.

The title of the work has been changed in such a way, and it is better to reflect the content of the work.

Ad 5.

This paper was translated from Polish to English by the native English speaker.

Ad. 6

The abbreviations have been spelled in full in revision of this article.

Reviewer 3 Report

The paper details some numerical and experimental analyses involving a shank with a thread and cooperating nut of an aircraft fuel system assembly. With reference to the manuscript title, however, the authors lack to provide further details on how the presented results are correlated with actual aircraft accidents mentioned in the title. Alternatively, it seems reasonable to consider FEM analyses as a well-established approach to design aeronautical structures following the applicable EASA regulations in aviation. A clear reference to the actual application of the case study would have been also beneficial. It is also recommended to better detail the main scientific achievements of the manuscript to consolidate the originality of the work.

Author Response

Reviewer 3

Thank you very much for the thorough review.

  1. This paper has been presented research into the cause of a military aircraft crash, and because of military secret, it is impossible to give any details of this accident.
  2. Regulations of EASA are not applicable in this particular research, because this investigation has been focused on military aircraft crash cause. In the air force technical authorities EASA regulations are nor applicable.
  3. This investigation has been focused on military aircraft crash cause. Due to the military secret, we cannot indicate on which aircraft described malfunction has occured, caused the fatal accident.

Round 2

Reviewer 2 Report

The paper has been substantially improved.

Add weaknesses / limitations of the study in the Conclusion.

Author Response

Dear Reviewer,

We made revisions in the Conclusion.

Thank you.